

# Validation and application of the Chinese version of the Perceived Stress Questionnaire (C-PSQ) in nursing students

Yi Luo[1],*, Boxiong Gong[2],*, Runtang Meng[3],*, Xiaoping Cao[1], Shuang Tang[4], Hongzhi Fang[5], Xing Zhao[6] and Bing Liu[7]

[1] School of Nursing, Ningbo College of Health Sciences, Ningbo, People's Republic of China
[2] Department of Oncology, Taihe Hospital, Hubei University of Medicine, Shiyan, People's Republic of China
[3] Department of Preventive Medicine, School of Health Sciences, Wuhan University, Wuhan, People's Republic of China
[4] Department of Nursing, Renmin Hospital, Hubei University of Medicine, Shiyan, People's Republic of China
[5] Department of Anaesthesiology, Shandong Provincial Western Hospital, Jinan, People's Republic of China
[6] Department of Orthopaedics, Taihe Hospital, Hubei University of Medicine, Shiyan, People's Republic of China
[7] Centre of Health Administration and Development Studies, Hubei University of Medicine, Shiyan, People's Republic of China
* These authors contributed equally to this work.

Corresponding author
Runtang Meng,
mengruntang@whu.edu.cn

## ABSTRACT

**Objective:** To translate the Perceived Stress Questionnaire (PSQ) into Chinese, validate its reliability and validity in nursing students and investigate the perceived stress level of nursing students.

**Method:** Forward- and back-translation combined with expert assessment and cross-cultural adaptations were used to construct the Chinese version of the PSQ (C-PSQ). This research adopted a stratified sampling method among 1,519 nursing students in 30 classes of Ningbo College of Health Sciences to assess the reliability and validity of the C-PSQ. Among them, we used the Recent C-PSQ (only the last month).

**Results:** The C-PSQ retained all 30 items of the original scale. Principal component analysis extracted five factors that explained 52.136% of the total variance. The S-CVI/Ave was 0.913. Concurrent validity was 0.525 and 0.567 for anxiety and depression respectively. The results of the confirmatory factor analysis were as follows: $\chi^2/df$ = 4.376, RMR = 0.023, GFI = 0.921, AGFI = 0.907, CFI = 0.916, RMSEA = 0.048, PNFI = 0.832, PGFI = 0.782, CN = 342 and AIC/CAIC = 0.809. The scale's Cronbach's alpha was 0.922, and Cronbach's $\alpha$ of each dimension was 0.899 (worries/tension), 0.821 (joy), 0.688 (overload), 0.703 (conflict), 0.523 (self-realization). The correlation coefficient between the first and second test, the first and third test and the second and third test was 0.725, 0.787 and 0.731, respectively. Mean values and distribution of overall PSQ index in nursing students was 0.399 ± 0.138. Different demographic factors were significantly associated with the perceived stress of nursing students.

**Conclusion:** The C-PSQ has an appropriate reliability and validity, which means that the scale can be used as a universal tool for psychosomatic studies. The perceived stress of nursing students was relatively high. Further studies are needed.

## INTRODUCTION

Nursing students experience a substantial amount of stress (*Al-Zayyat & Al-Gamal, 2014*; *Patterson, 2016*). These perceived stresses increase in the process of learning professional nursing knowledge (*Lamaurt et al., 2011*; *Levesque, 2015*). Their stress originates from daily life events, the rigorous study of theories, and nursing clinical practice. On the one hand, nursing students must spend plenty of time and energy learning complicated professional topics, which makes them feel isolated, helpless and nervous (*Yearwood & Riley, 2010*). On the other hand, clinical practice is performed in the hospital, which has a complicated environment (heavy workload, quick tempo, highly concentrated and intense competition) and depressed atmosphere (birth, senility, illness and death); nursing is a high-risk occupation in China. Nursing students can feel anxious, as they are constantly exposed to the sad emotions of the patients and their family members as well as fear of the risk of needle stick injuries (*Moscaritolo, 2009*; *Shearer & Davidhizar, 1998*). Moreover, nursing practice requires nursing students to possess a high medical and humanistic quality; nursing students can experience great stress while studying to meet these requirements because of their fear of lacking professional knowledge and skills (*Moridi, Khaledi & Valiee, 2014*; *Sheu, Lin & Hwang, 2002*).

For most Chinese students, stress also results from characteristics of the Chinese education system. Inequality exists in the allocation of educational resources, and the educational resource-utilization-rate is low (*Rong & Shi, 2001*). In addition, as a result of the rapid expansion of the Chinese educational system, graduates' employment rate has become lower than before, which is uncommon in the development of higher education worldwide (*Wen, 2005*). Nursing students in China experience substantial stress. They not only tolerate the stress from academic studies and clinical practice but also from the risk of failing to find a job.

Excessive stress has negative effects on nursing students, including psychological disorders, physiological diseases and social maladjustments. Research indicates that stress can significantly predict depressive symptoms, the prevalence of depression has reached 32.6% among college nursing students (*Chen et al., 2015*). Another study shows that nursing students have a much higher probability of committing suicide than other students (*Goetz, 1998*). Excessive stress can therefore seriously affect nursing students' mental health and can cause physical injury. Moreover, it has been shown that stress increases the incidence of ulcerative colitis, sleeping difficulties and fatigue syndrome, which means that stress has a negative influence on students' health (*Asencio-López et al., 2015*; *Levenstein et al., 2000*, *2015*; *Waqas et al., 2015*). Poor mental and general health may

not only lead to a low capacity to study and cope in students (*Beddoe & Murphy, 2004*) but also change students' determination to engage in nursing practice, which may have poor physio–psycho-social responses (*Chen & Hung, 2014*; *Watson et al., 2009*).

The problems mentioned above present many challenges to nursing students as well as nursing educators. Nursing educators can gradually relieve students' stress and negative emotions through effective measures when they detect the students' perceived stress and recognize their nervousness and anxiety (*Hamaideh, Al-Omari & Al-Modallal, 2016*). The current study adopted the Perceived Stress Questionnaire (PSQ) to investigate nursing students' perceived stress level. In 1993, Susan Levenstein developed the PSQ and published it in English and Italian; it has shown good reliability and validity. The PSQ has two forms—the General PSQ and Recent PSQ. The General form measures the perceived stress based on the subjects' feeling in the long run ("in general, during the last two years"), while the Recent form evaluates according to events that happened in only the last month ("during the last month") (*Levenstein et al., 1993*). Two forms of the PSQ differ only in the defined time range, and other content are identical. The scale has 30 items that cover seven-dimensions including harassment, overload, irritability, lack of joy, fatigue, worries and tension. In addition to the English and Italian versions, the scale has been translated into other languages including German (*Fliege et al., 2001*, *2005*; *Kocalevent et al., 2007*, *2011a*, *2011b*), French (*Consoli et al., 1996*), Spanish (*Montero-Marin et al., 2014*; *Sanz-Carrillo et al., 2002*), Swedish (*Bergdahl & Bergdahl, 2002*; *Rönnlund et al., 2015*), Norwegian (*Østerås, Sigmundsson & Haga, 2015*), Greek (*Karatza et al., 2014*) and Thai (*Ross et al., 2005*; *Wachirawat et al., 2003*). We preliminarily focus on versions that provide relatively complete psychometric characteristics.

The PSQ belongs to a universal scale (*Kocalevent et al., 2007*) which is commonly used to measure perceived stress, it can be applied to the medical field and other fields (*Levenstein et al., 1994*). It provided an effective scale for the current study, as it has been used previously to measure perceived stress in medical students (*Montero-Marin et al., 2014*). Universal as the scale is, it can be used to measure the perceived stress of not only nursing students, medical students and inpatients (*Fliege et al., 2005*) but also that of the entire medical staff, such as doctors, nurses and managers. The Perceived Stress Scale (PSS) is another earlier universal scale for measuring stress perception and is currently translated into near 30 language versions (by the end of 2017), including the Chinese PSS, other than English on the basis of Laboratory for the Study of Stress, Immunity and Disease. Indeed, the major difference between the PSS and the PSQ lies solely with measurement dimensions, dimensions of the latter are more focused on individuals appraise situations in their lives as stressful to report whether there seem to be unpredictable, uncontrollable or overloaded during the previous month (*Lee, 2012*; *Levenstein et al., 1993*). According to items, there are three versions of the PSS (PSS-14, PSS-10 and PSS-4).

However, no Chinese version of the PSQ had been published until we introduced the Chinese version of the PSQ (C-PSQ). The C-PSQ was validated in a large sample of Chinese nursing students to measure their level of perceived stress, thus proving the scale had an appropriate reliability and validity. Once the PSQ has been introduced to China,

people will be able to use it to measure the perceived stress level of nursing students and other medical students as well as that of medical workers and other groups of people whose level of perceived stress needs to be studied. We believed that the development of the C-PSQ would provide a firm foundation for related studies in China.

## METHOD

### Introducing the scale

The PSQ was translated using forward- and back-translation based on the integrated method (*Sidani et al., 2010*) and Brislin's translation model (*Brislin, 1970*; *Doris, Lee & Woo, 2003*) after receiving permission from the original author—Susan Levenstein. Firstly, forward translation was independently carried out by two bilingual translators whose first language was Chinese. One translator had abundant psychological knowledge and knew the scale, while the other translator was sensitive to expressions of language. Secondly, the translator with abundant psychological knowledge and an English scholar compared and examined the two scales together to finalize a draft. Thirdly, two English language scholars who knew nothing about the English version of the PSQ back-translated the draft to an English version. Fourthly, the two back-translated scales were compared, and the back-translated version was finalized. Fifthly, the researcher compared and judged the differences between the back-translated manuscript and the original scale, forward- and back-translated different items again and finalized the questionnaire. Additionally, we consulted 10 scholars who are experts in the development and validation of scales from Wuhan University, Yunnan University and Ningbo College of Health Sciences. Taking the experts' suggestions and the results of the forward and backward translation into consideration, we developed the C-PSQ after several rounds of discussion. For the specific processes, refer to Fig. 1.

The C-PSQ maintains the item order and scoring method of the original English version of the PSQ, using a four-point Likert Scale and asking how often (on a scale from 1, "almost never," to 4, "usually") each item occurred. The lowest score on the original scale is 30, and the highest score is 120. The final score, PSQ index, is (raw score-30)/90 and ranges from 0 to 1, with higher scores indicating greater stress. Several items (1, 7, 10, 13, 17, 21, 25, 29) were reverse scored (*Levenstein et al., 1993*). There are presently two ways to cut-off score concerning PSQ index evaluation. Two cut-off scores of the PSQ index were yielded in recent research by using the PSQ index mean score (M) and standard deviation (SD) of the population studied in order to divide the subjects into three groups, low level ($\leq M \pm SD$), moderate level ($>M \pm SD$ and $\leq M \pm 2SD$) and high level ($>M \pm 2SD$) of perceived stress (*Bergdahl & Bergdahl, 2002*; *Kocalevent et al., 2007*). Three cut-off scores of the PSQ index is divided according to quartile in earlier research (*Levenstein et al., 1993*; *Sanz-Carrillo et al., 2002*).

### Ethics statement

The medical ethics committee of Wuhan University School of Medicine (WUSM) approved this study. The current study adhered to the rules of the Declaration of Helsinki
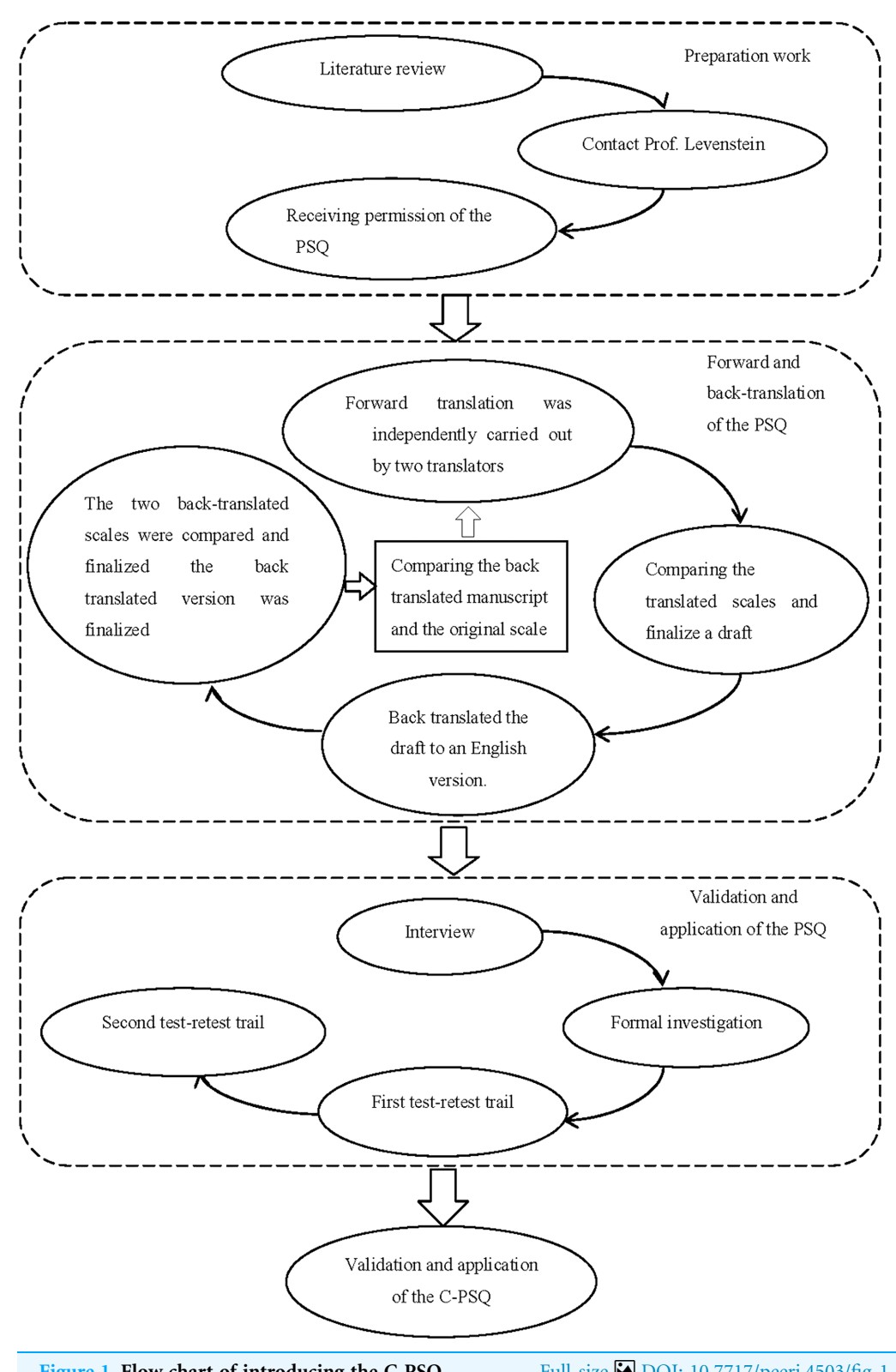

Figure 1 **Flow chart of introducing the C-PSQ.**

and its revised version as well as the rules of bio-medical journals. Additionally, this study was approved by the surveyed school and students in Ningbo College of Health Sciences.

## Research

The current study includes general information on and the perceived stress of nursing students. The newly developed C-PSQ was used to measure perceived stress. Among them, we used the Recent C-PSQ (only the last month). We interviewed nine students prior to conducting the survey formally to assess whether the general information form and language of the C-PSQ were suitable and reasonable in line with the Chinese context. We then revised the general information form based on the results of the interview and adjusted the text font, size and line spacing to make it easier to read to avoid information bias (*Althubaiti, 2016*).

The final general information form included the following information: sex, age, home location (city, town, village), single-child status, admission time (2015, 2014, 2013), initial educational degree (secondary school, high school), clinical practice experience, part-time job status, frequency of going back home, physical health, mental health, attitude toward nursing job prospects, greatest source of stress in college life (studies, employment, interpersonal relationships, love life, financial state, family), and the most often used coping skill (adjusting psychology, solving problems, escaping). Meanwhile, to test for criterion (concurrent) validity of the C-PSQ, the Goldberg Anxiety and Depression Scale (GADS, individually referred to as the GAS and GDS) was selected as a comparator scale, which shall be composed of a nine-item subscale that assesses symptoms of anxiety and a nine-item subscale that assesses symptoms of depression over the past month (*Goldberg et al., 1988*). All items can be answered with a simple "yes" or "no," with one or zero point respectively scored for each response. The final score is acquired by accumulating the response to each of the items, with higher values representing greater levels of symptomatology. The GADS has not only revealed good criterion validity for depressive disorders and generalized anxiety disorder but also displayed adequate values of sensitivity and specificity (*Kiely & Butterworth, 2015*; *Mulhall, Andel & Anstey, 2018*; *Pachana et al., 2007*). Our team used this brief and friendly scale because it has been widely adopted as a standard to screen of anxiety and depression in large sample studies of the general population (*Goldberg et al., 1988*).

The formal investigation occurred from November 18, 2015, to January 6, 2016. We adopted the stratified sampling method to identify the sample of nursing students in Ningbo College of Health Science. In total, 1,519 nursing students from 30 classes were surveyed. Among respondents, students in Grade 1 had studied nursing courses for more than three months, and students in Grade 3 had taken part in clinical practice in the hospital for more than two months. Simultaneously, we randomly chose a class to test the test–retest reliability of the C-PSQ. A total of 50 students in the class were tested three times including the formal survey, once per week; the final response rate was 100%. To fully respect and protect the subjects' privacy, subjects' responses to our study were considered anonymous and confidential. The objective of the survey and the instructions for filling out the form were explained to the nursing students before the survey was

conducted. All questionnaires were written and were collected once the subject finished writing. After excluding the unfinished and nonstandard questionnaires, a total of 1,453 complete questionnaires were collected, for a response rate of 95.66%.

## Statistical method

A database was built by Epidata (version 3.1; Lauritsen JM & Bruus M, Odense, Denmark) software. SPSS (version 18.0; SPSS Inc., Chicago, IL, USA), AMOS (version 18.0; SPSS Inc., Chicago, IL, USA) and Excel (version 2010; Microsoft, Redmond, WA, USA) were adopted to analyse the data. Descriptive statistics were used to describe the demographic characteristics. Construct validity was tested by factor analysis, which was performed using principle components analysis with varimax transformation. Criterion (concurrent) validity and convergence values were evaluated by Spearman's correlations. Ten experts evaluated the content validity of the scale and found it acceptable (*Lynn, 1986*). We chose Cronbach's coefficient to test the internal consistency of the scale and Spearman's correlations to assess the test–retest reliability. The (mean ± SD) represents the mean value; *T*-test or ANOVAs were used to compare the test or factor scores between two or more groups. The significance level was set at or below 5%.

## RESULTS

Characteristics of the subjects are described in Table 1. Nursing students' age ranged from 17 to 23 years, with an average age of 19.58 ± 1.09. Their length of clinical practice experienced was 2–12 months, and the average length was 8.58 ± 1.32 months.

The Kaiser–Meyer–Olkin measure of sampling adequacy (KMO) was 0.951, which means that the factor analysis was suitable (*Kaiser & Rice, 1974*). The common factors and component matrix of the principal component analysis are summarized in Table 2. The five extracted factors explained 52.136% of the total variance (>50%), which was an acceptable level (*Wu, 2010*). Factor 1 (worries/tension) includes 12 items (9, 12, 14, 15, 18, 19, 20, 22, 26, 27, 28, 30), factor 2 (joy) includes seven items (1, 10, 13, 17, 21, 25, 29), factor 3 (overload) includes four items (4, 8, 11, 16), factor 4 (conflict) includes five items (2, 3, 5, 6, 24) and factor 5 (self-realization) includes two items (7, 23). The five factors described below formed the five-dimensions of the scale.

Of the five extracted factors of the C-PSQ, items 2, 3, 5, 6 and 24 represented conflict, as they mainly revealed the socially acceptable degree of stress and psychological contradictions (*Rönnlund et al., 2015*; *Sanz-Carrillo et al., 2002*). Items 4, 8, 11 and 16 were named overload, as they mainly illustrated the stress caused by excess loads (*Levenstein et al., 1993*). Items 1, 10, 13, 17, 21, 25 and 29 were named joy, as they mainly presented a state that was joyful and energetic (*Sanz-Carrillo et al., 2002*); items on this dimension were reversely scored. Items 9, 12, 14, 15, 18, 19, 20, 22, 26, 27, 28 and 30 revealed the worry and strain of the subjective; as it combined the dimensions of worries and tension in the original scale, we named it worries/tension (*Levenstein et al., 1993*). Items 7 and 23 represented self-realization, and thus we called it self-realization (*Sanz-Carrillo et al., 2002*). We compared the scale's items clustering in the factors and factorial structure

**Table 1 Nursing students' demographic data (N = 1,453).**

|  | n (%) |
|---|---|
| **Sex** |  |
| Male | 20 (1.38) |
| Female | 1,433 (98.62) |
| **Age** |  |
| 17 | 8 (0.55) |
| 18 | 239 (16.45) |
| 19 | 457 (31.45) |
| 20 | 469 (32.28) |
| 21 | 209 (14.38) |
| 22 | 69 (4.75) |
| 23 | 2 (0.14) |
| **Home location** |  |
| City | 194 (13.35) |
| Town | 869 (59.81) |
| Village | 390 (26.84) |
| **Single-child status** |  |
| Yes | 473 (32.55) |
| No | 980 (67.45) |
| **Admission year** |  |
| 2015 | 603 (41.50) |
| 2014 | 566 (38.95) |
| 2013 | 284 (19.55) |
| **Clinical practice status** |  |
| Yes | 653 (44.94) |
| No | 800 (55.06) |

between the C-PSQ, English/Italian version of the PSQ and other versions of the PSQ. The results are shown in Table 3.

The average Content Validity Index of the PSQ (S-CVI/Ave) was 0.913 (>0.90), which means that the scale has good content validity (*Polit & Beck, 2006*). Taking the GADS as criterion, concurrent validity of the PSQ was 0.525 and 0.567 for anxiety and depression respectively. The results of construct validity of the PSQ displays in Table 4.

Based on the results of the factor analysis above, we conducted a confirmatory factor analysis to modify the model and formed Fig. 2. In addition, the uncorrelated base model demonstrates in Fig. 3. The results of the tests and the model's goodness of fit are shown in Table 5.

From Table 5, we could see that the model's chi-square degree of freedom was 4.376; comprehensive assessments were made by referring to the goodness-fit index, as the result could be influenced by sample size (*Kline, 2016*; *Wheaton, 1987*). Indices that were within the standard range included RMR = 0.023, GFI = 0.921, AGFI = 0.907, CFI = 0.916, RMSEA = 0.048, PNFI = 0.832, PGFI = 0.782, CN = 342 and AIC/CAIC = 0.809. The critical values for each of the fit indices (*Byrne, 2016*) are RMR <0.05, GFI >0.9

**Table 2 Communalities and rotated component matrix.**

|  | Communalities | Component | | | | |
|---|---|---|---|---|---|---|
|  |  | Factor 1 | Factor 2 | Factor 3 | Factor 4 | Factor 5 |
| a20 | 0.554 | **0.645** | 0.199 | −0.010 | 0.307 | 0.062 |
| a22 | 0.532 | **0.634** | 0.130 | 0.068 | 0.190 | 0.268 |
| a28 | 0.558 | **0.620** | 0.249 | 0.244 | 0.198 | −0.109 |
| a12 | 0.551 | **0.611** | 0.187 | 0.050 | 0.317 | 0.198 |
| a27 | 0.571 | **0.602** | 0.281 | 0.346 | 0.093 | −0.005 |
| a19 | 0.522 | **0.593** | 0.120 | 0.371 | 0.136 | 0.020 |
| a14 | 0.427 | **0.551** | 0.131 | 0.302 | 0.087 | 0.083 |
| a18 | 0.543 | **0.543** | 0.217 | 0.399 | 0.122 | 0.160 |
| a30 | 0.533 | **0.531** | 0.094 | 0.440 | −0.007 | 0.218 |
| a26 | 0.515 | **0.531** | 0.283 | 0.373 | 0.030 | 0.115 |
| a15 | 0.527 | **0.456** | 0.176 | 0.398 | 0.193 | 0.305 |
| a9 | 0.439 | **0.420** | 0.135 | 0.382 | 0.138 | 0.283 |
| a21 | 0.654 | 0.233 | **0.756** | −0.043 | 0.142 | 0.080 |
| a13 | 0.583 | 0.189 | **0.702** | 0.009 | 0.092 | 0.215 |
| a25 | 0.485 | 0.201 | **0.647** | 0.151 | −0.049 | 0.012 |
| a10 | 0.480 | 0.135 | **0.647** | 0.093 | 0.173 | 0.071 |
| a1 | 0.459 | 0.124 | **0.635** | 0.117 | 0.069 | 0.150 |
| a29 | 0.513 | 0.071 | **0.628** | 0.230 | −0.018 | −0.245 |
| a17 | 0.419 | 0.108 | **0.615** | −0.033 | 0.137 | 0.095 |
| a4 | 0.572 | 0.064 | 0.094 | **0.729** | 0.149 | 0.070 |
| a11 | 0.427 | 0.238 | −0.115 | **0.592** | 0.063 | −0.052 |
| a8 | 0.564 | 0.279 | 0.295 | **0.563** | 0.144 | 0.248 |
| a16 | 0.433 | 0.345 | 0.067 | **0.538** | 0.104 | 0.099 |
| a5 | 0.554 | 0.276 | 0.113 | 0.010 | **0.682** | 0.007 |
| a3 | 0.502 | 0.088 | 0.144 | 0.342 | **0.571** | 0.175 |
| a2 | 0.524 | 0.095 | 0.115 | 0.402 | **0.565** | −0.146 |
| a6 | 0.520 | 0.294 | 0.090 | 0.162 | **0.541** | 0.327 |
| a24 | 0.520 | 0.489 | 0.089 | −0.018 | **0.522** | −0.033 |
| a23 | 0.553 | 0.267 | 0.114 | 0.222 | 0.030 | **0.647** |
| a7 | 0.608 | 0.051 | 0.523 | 0.001 | 0.079 | **0.571** |

**Note:**
"a" Represents item. In bold are the highest loading for each item and loadings at 0.40 or higher.

(*Hu & Bentler, 1999*), AGFI >0.9, CFI >0.9 (*Bentler, 1990*; *Hu, Bentler & Hoyle, 1995*), RMSEA <0.05 (good fit) or <0.08 (reasonable) (*Browne & Cudeck, 1993*), PNFI >0.5, PGFI >0.5, CN >200 (*Hu, Bentler & Hoyle, 1995*), lower is better concerning AIC/CAIC value (*Wu, 2010*), respectively.

Cronbach's alpha of the C-PSQ was 0.922 CI [0.916–0.928], which means that this scale has good internal consistency (*Antonius, 2003*). Moreover, Cronbach's α values of the other five-dimensions were all acceptable (*Wu, 2010*), including 0.899 CI [0.891–0.907], 0.821 CI [0.807–0.835], 0.688 CI [0.661–0.713], 0.703 CI [0.678–0.726] and 0.523
**Table 3 Comparison of factorial structure among different versions of the PSQ.**

| Original version | Spanish version | German version | Greek version | Swedish version | Chinese version |
|---|---|---|---|---|---|
| Harassment (2, 6, 19, 24) | Harassment–social acceptance (5, 6, 12, 17, 19, 20, 24) | – | Harassment (6, 19, 24) | Conflict (6, 20, 24) | Conflict (2, 3, 5, 6, 24) |
| Overload (4, 11, 28, 29) | Overload (2, 4, 11, 18) | Demands (2, 4, 16, 29, 30) | Overload (2, 4, 11, 16, 18, 25, 28, 30) | Demand (2, 4, 11, 16, 29, 30) | Overload (4, 8, 11, 16) |
| Irritability (3, 10) | Irritability–tension–fatigue (1, 3, 8, 10, 14, 15, 16, 26, 27, 30) | – | | – | – |
| Lack of joy (5, 7, 16, 17, 21, 23, 25) | Energy–joy (1, 13, 21, 25, 29) | Joy (7, 13, 17, 21, 25) | Joy (1, 7, 13, 17, 21, 29) | Lack of joy (10, 17, 21, 25) | Joy (1, 10, 13, 17, 21, 25, 29) |
| Fatigue (1, 8, 13, 15) | – | – | Tension–fatigue (3, 5, 8, 10, 14, 26, 27) | Fatigue (1, 8, 13) | – |
| Worries (9, 18, 20, 22, 30) | Fear–anxiety (22, 28) | Worries (9, 12, 15, 18, 22) | Worries (9, 12, 15, 20, 22, 23) | | Worries/tension (9, 12, 14, 15, 18, 19, 20, 22, 26, 27, 28, 30) |
| Tension (12, 14, 26, 27) | – | Tension (1, 10, 14, 26, 27) | | Worries/tension (9, 14, 22, 27) | |
| – | Self-realization–satisfaction (7, 9, 23) | – | – | – | Self-realization (7, 23) |

**Note:**
C-PSQ (2017), the Greek version (2014) and the Spanish version (2002) keeps all the 30 items of the original version (1993) while the German version (2005) keeps 20 items and the Swedish version (2015) keeps 21 items of the original scale.

**Table 4 Convergence values for the C-PSQ hierarchical factors structure.**

| | rg | Mean | SD | Factor 1 | Factor 2 | Factor 3 | Factor 4 | Factor 5 |
|---|---|---|---|---|---|---|---|---|
| Perceived stress | 0–1 | 0.399 | 0.138 | 0.913 | 0.735 | 0.678 | 0.715 | 0.563 |
| Anxiety | 0–9 | 4.503 | 2.441 | 0.499 | 0.396 | 0.347 | 0.386 | 0.268 |
| Depression | 0–9 | 3.577 | 2.343 | 0.549 | 0.435 | 0.343 | 0.390 | 0.316 |

**Note:**
rg, range; SD, standard deviation; anxiety and depression from GADS; convergence values are Spearman's R correlations; all $P$ values are less than 0.01; correlation is significant at the 0.01 level (two-tailed).

CI [0.472–0.570]; namely 0.899 (worries/tension), 0.821 (joy), 0.688 (overload), 0.703 (conflict), 0.523 (self-realization). The scale has shown acceptable test–retest reliability. The correlation between the first and second test was 0.725 CI [0.514–0.878], the correlation between the first and third test was 0.787 CI [0.607–0.890] and the correlation between the second and third test was 0.731 CI [0.506–0.897]. These results at one-week intervals proved that the scale has an appropriate level of both stability and responsiveness to change over time. Reliability and validity of the PSQ in different nations show that in Table 6.

Mean values and distribution of overall perceived stress score (PSQ index) in the surveyed students was 0.399 ± 0.138 (0.02–0.90). By using the two cut-off scores described below, the prevalence of perceived stress at a moderate level was estimated to be 10.3%. The prevalence of perceived stress at high levels was 2.8%. Of the responding students,

**Table 5 Evaluation of the goodness of fit of the confirmatory factor analysis.**

| Index | Test result$^{\triangle}$ | Model fit judgement$^{\triangle}$ | Test result$^{\blacktriangle}$ | Model fit judgement$^{\blacktriangle}$ | Standard and critical value |
|---|---|---|---|---|---|
| $\chi^2/df$ | 4.376 | No (probably caused by the large sample) | 5.668 | No (probably caused by the large sample) | <3 |
| RMR | 0.023 | Yes | 0.030 | Yes | <0.05 |
| GFI | 0.921 | Yes | 0.896 | No | >0.9 |
| AGFI | 0.907 | Yes | 0.879 | No | >0.9 |
| CFI | 0.916 | Yes | 0.882 | No | >0.9 |
| RMSEA | 0.048 | Good fit | 0.057 | Reasonable | <0.05 (Good fit) <0.08 (Reasonable) |
| PNFI | 0.832 | Yes | 0.791 | Yes | >0.5 |
| PGFI | 0.782 | Yes | 0.771 | Yes | >0.5 |
| CN | 342 | Yes | 287 | Yes | >200 |
| AIC/CAIC | 0.809 | Relatively small | 0.854 | Relatively large | Relatively small |

Notes:
$\chi^2/df$, differences in chi-square by $df$ (all $P < 0.001$); RMR, root mean square residual; GFI, goodness-of-fit index; AGFI, adjusted goodness-of-fit index; CFI, comparative fit index; RMSEA, root mean square error of approximation; PNFI, parsimony-adjusted NFI; PGFI, parsimony goodness-of-fit index; CN, critical N; AIC, Akaike information criterion; CAIC, consistent Akaike information criterion.
$^{\triangle}$The modified model.
$^{\blacktriangle}$The uncorrelated base model.

647 (44.5%) thought that the greatest stress came from employment and 543 (37.4%) considered studying to be the greatest stress in college. Additionally, 49 students (3.4%) attributed the greatest stress to love affairs, while 50 students (3.4%) reported their financial situations. Eleven students (0.8%) ascribed stress to other categories. We compared the perceived stress of nursing students with different characteristics (Table 7).

## DISCUSSION

In the present study, the PSQ was translated and validated as well as applied in a large sample of nursing students. During the test–retest trial, the surveyed students reported engagement in different activities, including taking courses, skills training, sectional examinations and internship assignments. In particular, students were stressed during examinations and obtaining an internship, which we thought could influence their perceived stress and affect the final results. However, the results of the test–retest reliability were above 0.70, which meant that the scale are acceptable for research tools (*Keszei, Novak & Streiner, 2010*) and had certain stability. The concurrent validity and construct validity of the PSQ is not bad. Nonetheless, this result did not study using the same criterion as a reference. Therefore, the C-PSQ has an appropriate reliability and validity, which guarantees it as a suitable tool to measure the perceived stress of people in China.

$\chi^2/df$ can be influenced by sample size, which was large in the current study. As a result, the $\chi^2/df$ did not reach the appropriate standard (*Hayduk, 1987*), but the results were acceptable, as they matched the flexible range (<5) (*Wu, 2009*). Moreover, other goodness-of-fit indexes of the model were all within the acceptable range, demonstrating that the scale's structure was stable.

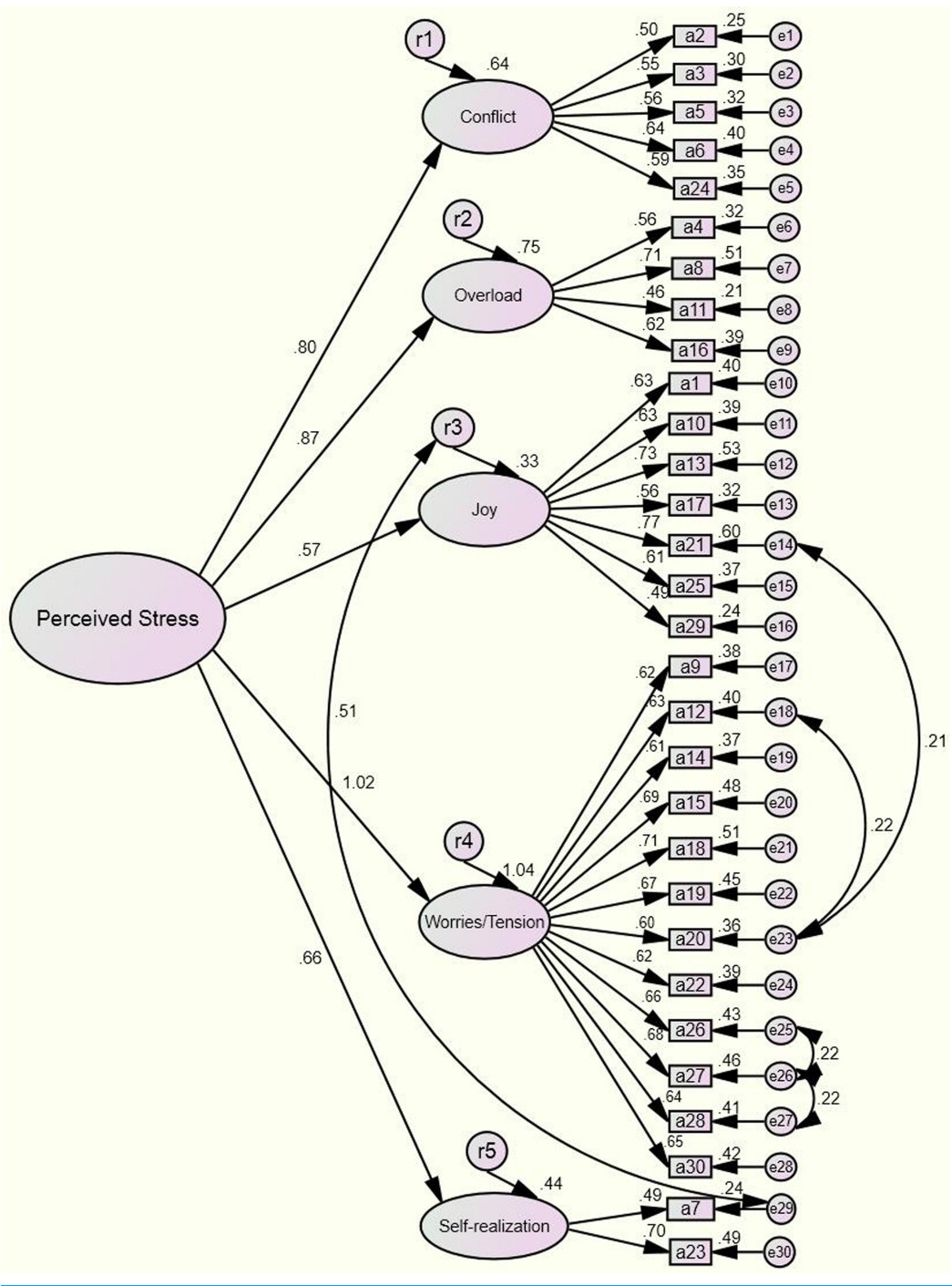

**Figure 2 Confirmatory factor analysis the modified model (*n* = 1,453).**

As shown in Table 3, visible difference in the structure and items clustering in the factors are present among different versions of the PSQ but on some level several items of the PSQ (24, 4, 21, 14 and 27) were happened to the cluster on a stability factor. In spite of this, the PSQ could be still translated into different languages and applied globally.

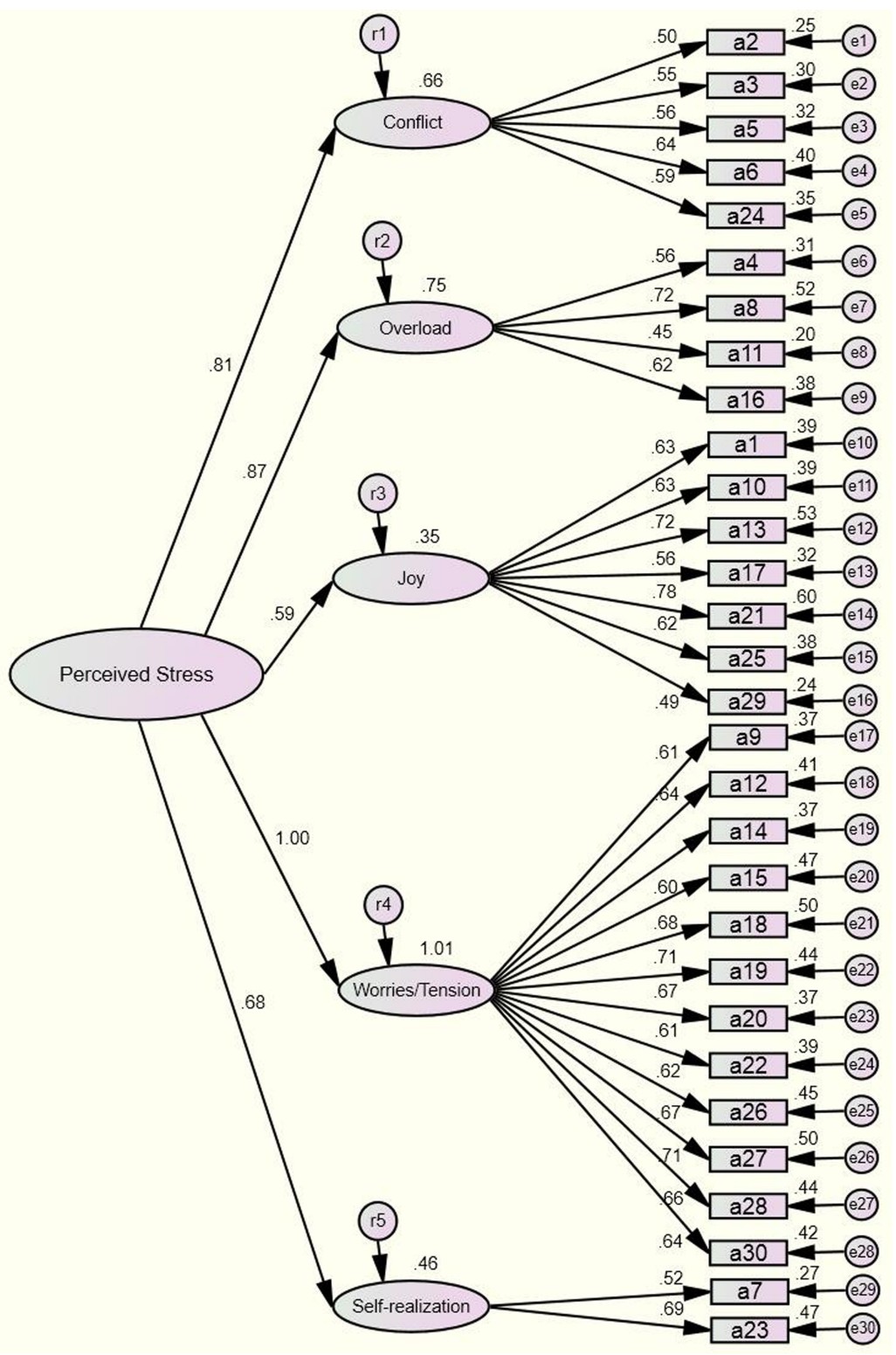

**Figure 3 Confirmatory factor analysis the uncorrelated base model (*n* = 1,453).**

**Table 6 Description on different validity and reliability of the PSQ reported in original as well as different translated versions.**

| | Original version | Spanish version | German version | Greek version | Swedish version | Chinese version |
|---|---|---|---|---|---|---|
| **Validity** | | | | | | |
| Criterion validity | 1. Trait anxiety (STAI) ($N = 24$): $r = 0.69^G$; $r = 0.75^R$<br>2. PSS ($N = 89$): $r = 0.56^G$; $r = 0.73^R$<br>3. CES-D ($N = 24$): $r = 0.49^G$; $r = 0.56^R$<br>4. Self-rated stress ($N = 52$): $r = 0.40^G$; $r = 0.56^R$<br>5. Somatic complaints (from Kellner's SQ) ($N = 73$): $r = 0.50^G$; $r = 0.58^R$ | 1. Trait anxiety (STAI) ($N = 80$): $r = 0.65^G$; $r = 0.69^R$<br>2. Depression (BDI) ($N = 80$): $r = 0.46^G$; $r = 0.49^R$<br>3. Psychological disturbance (GHQ-28) ($N = 80$): $r = 0.51^G$; $r = 0.55^R$<br>4. Somatic symptoms (SPPI somatic section) ($N = 31$): $r = 0.62^G$; $r = 0.67^R$ | 1. Quality of life (WHOQoL): $r = -0.58^G$<br>2. Social support (F-SOZU): $r = -0.61^G$<br>3. Higher perceived stress scores are associated with some of the relevant indicators of a supposed immunological imbalance (tryptase$^+$ mast cells (TMC$^+$), CD8$^+$ T-cells and TNF-$\alpha^+$ cells) in women who have had a miscarriage | 1. DASS-21 ($N = 451$): $r = 0.597^R$<br>2. PSS ($N = 453$): $r = 0.737^R$ | 1. BDI ($N = 1,275$): $r = 0.354^G$<br>2. State anxiety (STAI) ($N = 1,275$): $r = 0.400^G$<br>3. Trait anxiety (STAI) ($N = 1,275$): $r = 0.539^G$ | 1. GAS ($N = 1,453$): $r = 0.525^R$<br>2. GDS ($N = 1,453$): $r = 0.567^R$ |
| Content validity | Null | Null | Null | 0.2–0.5 (inter-correlations between the items) | Null | 0.913 (S-CVI/Ave) |
| Construct validity | Seven factors 60%* | Six factors 58.01%* | Four factors 58%* | Five factors 54.28%* | Five factors 55.5%* | Five factors 52.136%* |
| **Reliability** | | | | | | |
| Coefficient alpha | $0.90^G$; $0.92^R$ | $0.90^G$; $0.87^R$ | At least $0.85^G$ (reliability at least 0.80) | $0.90^R$ | $0.90^G$ | $0.922^R$ |
| Test–retest | $0.82^G$ $8.03 \pm 1.64$ days, $N = 101$ | $0.80^G$ $13.12 \pm 2.05$ days, $N = 176$ | Null | $0.86^R$ One month, $N = 212$ | Null | $0.725^R$ One week, $N = 50$ |

**Notes:**
1. STAI, the state–trait anxiety inventory; PSS, the perceived stress scale; CES-D, the Center for Epidemiologic Studies Depression; BDI, the beck depression inventory; GHQ-28, the General Health Questionnaire-28 items; DASS-21, the Depression Anxiety Stress Scale (a short version); GADS = GAS + GDS, the Goldberg Anxiety and Depression Scale.
2. G represents the General PSQ; R means the Recent PSQ.
* Cumulative variance.

**Table 7 Comparison of perceived stress in nursing students.**

|  | n | Mean ± SD | t/F | P |
|---|---|---|---|---|
| Initial educational degree |  |  | −9.749 | 0.000 |
| Secondary school | 319 | 0.334 ± 0.123 |  |  |
| High school | 1,134 | 0.417 ± 0.136 |  |  |
| Clinical practice |  |  | −8.823 | 0.000 |
| Yes | 653 | 0.364 ± 0.127 |  |  |
| No | 800 | 0.427 ± 0.140 |  |  |
| Part time job |  |  | 4.816 | 0.000 |
| Yes | 570 | 0.420 ± 0.140 |  |  |
| No | 883 | 0.385 ± 0.135 |  |  |
| Frequency of going home |  |  | 5.348 | 0.001 |
| <1/2 Month | 311 | 0.380 ± 0.130 |  |  |
| <1 Month | 465 | 0.390 ± 0.139 |  |  |
| <1 Season | 305 | 0.410 ± 0.134 |  |  |
| <1 Semester | 372 | 0.417 ± 0.143 |  |  |
| Physical health |  |  | 69.537 | 0.000 |
| Very good | 334 | 0.336 ± 0.126 |  |  |
| Good | 779 | 0.393 ± 0.126 |  |  |
| Average | 316 | 0.469 ± 0.137 |  |  |
| Bad | 24 | 0.551 ± 0.156 |  |  |
| Mental health |  |  | 134.761 | 0.000 |
| Very good | 391 | 0.324 ± 0.124 |  |  |
| Good | 737 | 0.394 ± 0.118 |  |  |
| Average | 300 | 0.489 ± 0.128 |  |  |
| Bad | 25 | 0.631 ± 0.134 |  |  |
| Prospect of employment |  |  | 45.702 | 0.000 |
| Very good | 106 | 0.325 ± 0.139 |  |  |
| Good | 683 | 0.373 ± 0.127 |  |  |
| Average | 608 | 0.431 ± 0.136 |  |  |
| Bad | 56 | 0.512 ± 0.133 |  |  |
| Coping skill |  |  | 48.516 | 0.000 |
| Adjusting psychology | 968 | 0.388 ± 0.131 |  |  |
| Solving problems | 369 | 0.390 ± 0.138 |  |  |
| Escaping | 116 | 0.516 ± 0.138 |  |  |

Note:
Secondary school and high school represent the educational degree before college degree. $t/F$, we chose "$t$" to compare the differences between the two groups; we used "$F$" to compare differences between more than two groups.

Most of the fit statistics of the modified model is greater than the critical value and around half of fit statistics of the uncorrelated base model are not satisfactory in this study. We must admit that no matter which model's fitting effect is not great satisfactory, the modified model we reluctantly accept. Multi-country study showed that the results of exploratory factor analysis are inconsistent after the PSQ was translated into local languages. There is reason to believe that the structural equation model may need to be further simplified.

We will consider removing items (item reduction) (*Fliege et al., 2005*; *Rönnlund et al., 2015*) to optimize the structure of factors in future studies.

Furthermore, based on appropriate reliability and validity, we retained all 30 items of the original scale (*Levenstein et al., 1993*), thereby maintaining the high integrity of the original scale in obtaining an objective result. Moreover, the original English and Italian scales had advanced after 20 years of development, and items of the C-PSQ kept the same items as the original scale as well as the item order (*Asencio-López et al., 2015*; *Levenstein et al., 1993*, *1994*, *2000*). Including reversed scores for some of the items can detect false information. For example, when a subject chose "usually" as the answer for both "you feel rested" and "you feel tired," we judged the response as ineffective. In word, we need to extend the sample further research concerning reliability and validity of the PSQ.

Mean values and distribution of overall PSQ index in nursing students was $0.399 \pm 0.138$. This index was lower than that of ulcerative colitis patients in Susan Levenstein's research (*Levenstein et al., 1994*). Independent $t$-tests revealed that the differences were not statistical significant, $t = -1.659$, $P = 0.097$. This index was higher than that of the general population (*Sanz-Carrillo et al., 2002*), $t = 4.024$, $P = 0.000$, and this difference was statistically significant. In the current study, nursing students' perceived stress levels were relatively high, which was consistent with the results of other studies (*Lee & Noh, 2016*; *Ross et al., 2005*). Appropriate stress can motivate students' enthusiasm to study and practice and can cultivate their confidence and optimism. However, students are forced to cope with stress when it becomes excessive (*Findik et al., 2015*). Whether the stress results in unhealthy physical and psychological change or abnormal behavior depends on factors such as social support (school, family, friends and community) (*Metzger et al., 2016*) and self-adjustment (*Saoji, 2016*). Therefore, it is necessary for nursing educators to recognize nursing students' stress and communicate with students to gradually build a support system for them. Leading and encouraging the students to develop mechanisms that facilitate optimism can help students manage stress and stay in a good mood.

Students whose initial educational degree was secondary school had lower perceived stress levels than students whose initial educational degree was high school. This could be explained by the previous nursing experience gained by secondary school graduates during their schooling. They became accustomed to the nursing field earlier than students who directly graduated from high school, and as students who directly graduated from high school were unfamiliar with the study of nursing, they became stressed. Moreover, students who participated in clinical practice had a lower perceived stress than those who did not; this result differed from other studies (*Al-Zayyat & Al-Gamal, 2014*; *Moridi, Khaledi & Valiee, 2014*). Traditionally, people think that clinical practice is the greatest source of stress for nursing students. We speculated that students' perceived stress originated most from their fear of the many uncertain events that could happen during their internship, rather than their involvement in clinical practice. Students who are about to participate in their internship had a higher perceived stress, as they were worried and feared the difficulties they might face, whereas students who had participated in the internship had a lower perceived stress, as they were able to accomplish their work.

Part-time jobs influenced nursing students in many ways (*Lee, Mawdsley & Rangeley, 1999*). Students who worked part-time were under greater stress than those who did not. This might be because students who take part-time positions have a heavier economic burden than those who do not; they have to make a living through this work (*Warning over nursing students who resort to part-time jobs just to get by, 2015*). Moreover, role conflicts occur when students play many roles in their life, including student, worker and friend (*Yamada et al., 2011*). Studies show that time spent on part-time position is inversely proportional to students' scores. Working 16 or more hours per week has a negative influence on students' academic achievements (*Salamonson & Andrew, 2006*). Working students' learning schedules could be occupied by their part-time job, thus leading to high levels of stress in studies and daily life.

Students who visited their home frequently had a lower perceived stress level than those who did not. Going back home can comfort nursing students through the provision of family support. One study showed that family support played an important role in medical students' life, especially when they were faced with a challenge. Family support encouraged students to face that challenge head-on and full of confidence (*Klink, Byars-Winston & Bakken, 2008*). Furthermore, it can affect students' anxiety and depression (*Wodka & Barakat, 2007*), lower the incidence of depression (*Harris & Molock, 2000*) and positively affect the psychological health of students.

Students who were optimistic about their employment had a lower perceived stress than those who were not. Employment stress is determined by both inward and outward influencing factors and is closely related to the environment, physiology, psychology and behavior (*Hwang, 2012*; *Yun & Kim, 2012*). For instance, stress in academics and daily life can cause students to lack confidence and determination when needing to find employment. Additionally, in recent years, the job market has been stressful, which presents a challenge to Chinese nursing students.

Furthermore, students who could manage their emotions and were good at solving problems had a lower perceived stress than those who tended to avoid stress. One of the keys to success is knowing how to cope with stress and difficulties (*Brady et al., 2016*). Positive psychological interventions can be useful in reducing stress and improving confidence (*Greeson, Toohey & Pearce, 2015*; *Heinen, Bullinger & Kocalevent, 2017*). One strategy to improve health status is promoting stress management capacity through training (*Li et al., 2016*). One study showed that rational coping strategies were inversely proportional to perceived stress (*Crego et al., 2016*). Moreover, the students who were psychologically and physiological healthy had lower perceived stress levels than those who were not. Students' perceived stress can both influence and be influenced by their psychological and physiological health. Further studies should be conducted on the process of how stress influences psychological and physiological health.

## CONCLUSION

The C-PSQ has an appropriate reliability and validity, which means that the scale can be used as a universal tool for psychosomatic studies. The perceived stress of nursing

students was relatively high. In future research, it is necessary to further expand the sample to test different groups. Further studies are needed.

## Relevance for clinical practice

The current study has translated the PSQ into Chinese and applied it to nursing students. Results showed that nursing students' perceived stress level was relatively high which remind nursing educators to focus on students' stress. High level of stress makes students give up nursing study, educators should avoid this phenomenon which may cause the loss of clinical nurse and influence the nursing service quality. Furthermore, the PSQ could also be applied to clinical nurses by which the nursing managers could know the perceived stress of nurses. Nursing managers would relieve the stress of nurses which can ensure the smooth development of nursing work. We suggested that future studies should continuously monitor the dynamic stress level of nurses throughout their nursing career, specific interventions would be made in some special time of nodes at which the stress level is high. Such interventions would promote the development of nurses and improve the stability of the nursing team.

### Limitations and suggestions for future research

Despite our efforts to completely explore validation and application of the C-PSQ, we firmly believe that additional psychometrics indicators and influencing factors should be incorporated into further research done in the future.

1. Validation should include construct validity, criterion validity and content validity tests. There will be critical need also for action to find more evidence to prove that validity of the C-PSQ has stable and good validity. There are no adequate comparator scales to establish criterion validity and construct validity of the scale being assessed. The PSS may be a suitable criterion for testing in future studies.
2. The cross-sectional design of this study only tested nursing students, resulting in limited the inference of application range. As the PSQ is a universal scale, we need to measure different samples of more locations to confirm the C-PSQ applicability in China.
3. The PSQ belongs to a subjective measurement scale with respect to stress perception, which is easily affected by various factors, such as participants' cultural level and participation attitude. If further studies can be combined with objective indicators (physiological and biochemical index) as a criterion, thereby obtaining a more comprehensive criterion-related validity.

## ACKNOWLEDGEMENTS

Special thanks to Susan Levenstein MD from Aventino Medical group in Italy and Chua Yeewen from Department of Psychology, HELP University for their great help in the process of introducing the scale to China. We really appreciate that Susan Levenstein MD has taken time out of her schedule to comment on this paper. Meanwhile, we would like to acknowledge our friends Jingjing Li PhD Candidate from Rollins School of Public Health (RSPH), Emory University; Zhenkun Wang PhD from Tongji Hospital, Tongji Medical

College, Huazhong University of Science and Technology; Yucong Ma who is a MTI of Southeast University–Monash University Joint Graduate School (Suzhou) and Di Zhang who is a master student from School of Health Sciences, Wuhan University for their valuable assist in forward and back-translation. Also, we are grateful to all subjects taking part in the present study and teachers in Ningbo College of Health Sciences for their necessary assistance in collecting data. All authors appreciated the reviewers for constructive comments and suggestions to improve the quality of manuscript.

### Funding

This project was supported by Key Research Center for Humanities and Social Sciences in Hubei Province (Hubei University of Medicine) (Grant No. 201612). The funders had no role in study design, data collection and analysis, decision to publish, or preparation of the manuscript.

### Grant Disclosures

The following grant information was disclosed by the authors:
Key Research Center for Humanities and Social Sciences in Hubei Province (Hubei University of Medicine): 201612.

### Competing Interests

The authors declare that they have no competing interests.

### Author Contributions

- Yi Luo conceived and designed the experiments, performed the experiments, analyzed the data, prepared figures and/or tables, authored or reviewed drafts of the paper, approved the final draft, the first draft was written by Yi Luo and Boxiong Gong jointly.
- Boxiong Gong prepared figures and/or tables, authored or reviewed drafts of the paper, approved the final draft, the first draft was written by Yi Luo and Boxiong Gong jointly.
- Runtang Meng conceived and designed the experiments, performed the experiments, analyzed the data, contributed reagents/materials/analysis tools, prepared figures and/or tables, authored or reviewed drafts of the paper, approved the final draft, was mainly responsible for the revised manuscript.
- Xiaoping Cao performed the experiments, prepared figures and/or tables, authored or reviewed drafts of the paper, approved the final draft.
- Shuang Tang prepared figures and/or tables, authored or reviewed drafts of the paper, approved the final draft.
- Hongzhi Fang prepared figures and/or tables, authored or reviewed drafts of the paper, approved the final draft.
- Xing Zhao prepared figures and/or tables, authored or reviewed drafts of the paper, approved the final draft.
- Bing Liu prepared figures and/or tables, authored or reviewed drafts of the paper, approved the final draft.

## Human Ethics

The following information was supplied relating to ethical approvals (i.e., approving body and any reference numbers):

The medical ethics committee of Wuhan University School of Medicine (WUSM) approved this study. The current study adhered to the rules of the Declaration of Helsinki and its revised version as well as the rules of bio-medical journals. Additionally, this study was approved by the surveyed school and students in Ningbo College of Health Sciences.

## Data Availability

The raw data has been supplied as Supplemental Dataset Files.

## Supplemental Information

Supplemental information for this article can be found online at http://dx.doi.org/10.7717/peerj.4503#supplemental-information.

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
