# Peer review of "Validation and application of the Chinese version of the Perceived Stress Questionnaire (C-PSQ) in nursing students"

_PeerJ, doi:10.7717/peerj.4503_

## Round 0.1 · original submission · Major Revisions

· Academic Editor

Major Revisions

Dear authors,

The external reviewers have indicated scientific merit in your paper, but before its publication there are several importants changes you should apply. Therefore, you should re-send another version of the text with the indicated recommendations (MAJOR REVISION).

With respect and warm regards,
Dr Palazón-Bru (academic editor for PeerJ)

·

Basic reporting

The manuscript requires through revision to conform with academic writing style. Specific comments are provided in the general comments section.

Experimental design

Design have some concerns, for example, probable bias in back-translation procedure, lack of validity measures. Specific comments are provided in the general comments section.

Validity of the findings

Data need to be checked again however, further details on methods and procedures of analysis used by the authors are required.

Additional comments

Considering the extensive amount of data, I would suggest the authors to separately present validation (?) and application as two sub study (with a split sample) in this article. With detailed analyses and separate discussions that could be more useful for the readers.
The following section presents a few specific comments, however, these are only examples, when revising please check the manuscript for similar problems.
1. The conclusion written in the abstract section seems inappropriate. Please revise.
2. The CI values can be removed from the "Abstract" section.
3. Line 39, 273: Check in-text citation, some characters are missing.
4. The manuscript requires formatting. For example, parentheses of many in-text citations are not separated by space after the last world immediately before it (line 44, 48).
5. Line 45-46: It would be better to describe the "complicated environment" and "depressed atmosphere" instead of using these in their commonsense meaning (which may mean different things to different readers).
6. Line 60-61: Please rephrase "stress-caused depression".
7. Exaggerated claim: please rephrase these
a. irreparable damage (line 64)
b. can only (line72);
c. ideal universal scale (line 87).
8. It would be a good idea to discuss other available scales for measuring perceived stress scale (such as the "Perceived Stress Scale" developed by Cohen) in the introduction section.
9. Language correction required: line 81 - "content is" need to be "contents are"; line 88-90;
10. Please add description/statistics on different validity and reliability of the PSQ 30 reported in original as well as different translated versions.
11. Line 77: It sounds like two ways of using this questionnaire (general and recent) instead of two versions. Please clarify and revise subsequent sections as required.
12. Line 85-86: It is not clear what the authors meant to express with the last clause of this sentence. It seems that this sentence is unnecessary (in its current form).
13. Section 2.1: In the current form of paragraph, Firstly, Secondly, . . . . would fit better compared to First, Second, . . . ..
14. Line 122: It is crucial that the comparison between original and back-translation version of the scale is done by a person outside the research team. The possibility of bias may undermine the value of the entire translation process. Please justify the reason for using the research team members in this. Clarify how this might have not caused any negative impact on the confidence regarding similarity of the translation and original version.
15. The flow chart provided as Figure 1 needs to be parsimonious.
16. Line 123-124: The word "points" is misleading and may be deleted.
17. Line 126-129: How was the norm developed for the Spanish version? and on which population? It is not clear how that norm can be considered useful for (1)Chinese (2) Nursing students. Please justify the reason for using the Spanish norm for interpretation.
18. Line 140: What is meant by "normative" is unclear.
Line 140-141: If you want to use this sentence, please clarify how you tested if the "type setting and printing of the scale was standardized".
19. Line 159: Please clarify how you matched the test-retest data sheets as the responses were anonymous?
20. Line 158: As the ethics committee has approved this, I believe possibilities of unequal relation between participants and administrator/researcher should not be a concern from my side. However, a 100% response rate is very high considering such large number of student participants in this study.
21. Line 167: It is not clear why excel was used where the analysis used can be done with SPSS and AMOS.
22. Line 174: revise to avoid redundancy.
23. Table 1: Explain the purpose of grouping age in this manner (17-20 and 20-23).
24. Note below Table 1 is redundant.
25. Figure 2: It is unconventional to correlate error terms of different factors. Please explain the considerations based on which these were done. Please also compare the uncorrelated base model with the modified model. Discuss the influence of this modification in interpreting the findings.
26. Provide citation for the critical values for each of the fit indices used in the Table 4.
27. Line 216: Name of the dimensions (or factor number) need to be reported aside respective Cronbach's alpha values. An alpha value of 0.526 is not "acceptable" however, considering the low number of items (2; I assume it is Factor 5) it is understandable to have a low alpha.
28. Line 241: The authors' claim of C-PSQ to have good validity is unjustified and should be revised. There are several important types of validation including construct validity and concurrent validity which were missing in this research.
29. Line 247-249: Visible difference in the structure and items clustering in the factors are present among different versions of PSQ (Table 3). Therefore, authors' claims regarding "similarity" and high "structural stability" can be debatable and require revision.
30. Supplementary materials are in Chinese language, which need to translated for review.
31. Please provide further details on the data collection procedure (one to one/group, self administered/interviewer administered, who collected that data, who took consent and how).
32. Please provide details of the method used for factor analysis, my quick analysis with the provided data revealed some difference.

·

Basic reporting

Basic reporting in this study is adequate.

Experimental design

Experimental design has been well defined.

Validity of the findings

This study has adequate analysis and interpretation. However, a few additions would have made the study design better:

a) There are no comparator scales to establish criterion and concurrent validity of the scale being assessed. Please, list it as a limitation of this study.
b) It would be interesting to make a table providing reliability and validity of this scale and different factor structures arising in different nations.
c) If different factor structures have been reported in different nations, it would interesting to compare them in this study sample using AMOS software and see which one has the best fit. AIC statistic can be an important measure in comparing them. Please refer to this paper to adopt their methods: https://www.ncbi.nlm.nih.gov/pubmed/28815479

Additional comments

I would like to congratulate the authors on conducting this study of public health importance. However, there are few comments that should be addressed before

a) This statement should move to discussion: This score was lower than that of
ulcerative colitis patients in Susan Levenstein’s research(Levenstein et al. 1994).
b) The manuscript can benefit from improvement of English by a professional Anglophone editor.

Best wishes,

---

## Round 0.2 · accepted · Accept

· Academic Editor

Accept

Dear authors,

All your responses to the previous comments are valid, therefore I am happy to inform you that your paper has been accepted for publication in PeerJ.

Congratulations!

With respect and warm regards,
Dr Palazón-Bru (academic editor for PeerJ)

·

Basic reporting

No comment

Experimental design

No comment

Validity of the findings

No comment